# Study on Eccentric Uncoupled Blasting Effect of Cutting Seam Pipe

**Wei Wang, Jiaqi Zhang * and Ao Liu**

School of Civil Engineering, Central South University, No. 22 Shaoshan South Road, Changsha 410075, China; wangweicsu@csu.edu.cn (W.W.); 194812315@csu.edu.cn (A.L.)
* Correspondence: zjq.csu@csu.edu.cn

**Abstract:** In order to study the blasting effect and the damage to the rock mass when the cutting seam cartridge is eccentrically and uncoupled. The ANSYS/LS-DYNA® nonlinear dynamic platform was used to simulate the blasting effect of five eccentric uncoupled coefficients on the cutting seam cartridge, and the crack growth process under the condition of complete eccentricity was simulated. By comparing and analyzing the stress of measuring points in the direction of cutting seam, vertical cutting seam direction, and circumferential cutting seam pipe under different working conditions. It is concluded that the effect of detonation products is affected by the wrapping property of the cutting seam pipe and the eccentric uncoupled coefficient. With the increase of the eccentric uncoupled coefficient, the load distribution presents obvious non-uniformity. The pressure on the uncoupled side of the blasthole is smaller than that on the coupled side, and the peak time of the uncoupled side also lags behind that on the coupled side. When the eccentric uncoupled coefficient is 1, the peak pressure on the coupled side is 5.78 times that of the uncoupled side, and the explosive stress field is biased toward the coupled side. The existence of the cutting seam pipe causes stress concentration at the opening, which enhances the guiding effect of the initial crack, and the stress in the non-cutting seam direction is buffered. Therefore, the eccentric arrangement of the cutting seam pipe determines the formation of the initial crack and the subsequent blasting effect. When the cutting seam cartridge is arranged eccentrically and uncoupled, it will cause under-excavation at the connection direction of blasthole, which will cause less disturbance to the rock mass on the uncoupled side. If the retaining side rock mass is on the coupled side in actual blasting, the eccentric uncoupled arrangement will cause greater over-excavation and damage. Therefore, it is necessary to avoid this situation as far as possible and provide better guidance for the actual construction.

**Keywords:** cutting seam cartridge; directional fracture blasting; eccentric uncoupled coefficient; blasting effect

## 1. Introduction

In tunnel blasting and urban underground engineering, smooth blasting and pre-split blasting are usually used as control blasting techniques. However, to reduce the damage of blasting to rock mass, reduce the support cost, and ensure the quality of the tunnel forming, the methods such as the directional fracture blasting technology of the cutting seam cartridge and the uncoupled charging [1,2] are adopted to ensure the accurate control the direction of the explosion crack growth, effectively reduce the over-excavation and under-excavation, and improve the forming quality around the tunnel [3,4].

The essence of the cutting seam cartridge is to make use of the hard pipe with the axial slotted to make the detonation product act on the specific position of the blasthole wall first and precisely form the initial guiding crack, to meet the requirements of the flatness of blasting profile in directional blasting [5,6]. Since its proposal, it has been widely used in slope excavation, blasting around shafts, tunnel excavation, and other projects [7,8]. To avoid the excessive fragmentation of the local rock mass caused by the traditional coupled

charge form [9], it will affect the directional blasting effect. The cutting seam cartridge adopts an uncoupled charge structure, which can change the loading rate of the stress wave on the blasthole wall, prolong the action time of the stress wave and gas, increase the energy utilization rate of the explosive and improve the fragmentation distribution of blasting, which is more suitable for fracture forming controlled blasting technology such as directional fracture controlled blasting [10,11].

Regarding the cutting seam cartridge blasting technology, a large number of experts and scholars have studied the influence of the material of the cutting seam pipe [12], the shape of the groove [13], the way of grooving [14], the charge dose [15] and the uncoupled coefficient [16,17] on the blasting mechanism and crack propagation law. Cho et al. [18] used analysis software to simulate the crack growth law under different cutting seam cartridge blasting conditions. Yue, Z.W. et al. [19,20] studied the dynamic response effect of air-deck charge with cutting seam cartridge in cement mortar model test by using the super dynamic strain test system and concluded that the blasting effect is relatively ideal when the axial uncoupled coefficient is 1.5~2.0. Wang [21] combined dynamic caustics experiment and numerical simulation analysis to compare and study the blasting effect of the cutting seam cartridge when the air medium and the plasticine medium are filled between the explosive and the blasthole wall respectively. Wei, C.H. et al. [22] carried out a simulation study on the evolution of cracks generated by the cutting seam cartridge blasting under different ground stress conditions and analyzed the influence of different ground stress conditions on the blasting effect of the cutting seam cartridge. Yang, G.L. et al. [23], Yang, R.S. et al. [24] analyzed the dynamic fracture effect of detonation crack in the cutting seam cartridge blasting and concluded that the blasting effect was the most ideal when the uncoupled coefficient of the cutting seam cartridge blasting is 1.67. Wang, Y.B. et al. [25] studied the change of the stress intensity factor at the crack tip when the two blastholes are simultaneously initiated under different cutting seam methods.

Most of the above studies on the blasting effect and mechanism of the cutting seam cartridge blasting are conducted by model test [26], theoretical analysis [27,28], and other methods. However, the model experiment is usually in an extreme environment of high temperature and high pressure, so the whole process of the blasting, cannot be observed intuitively. The digital laser dynamic caustics test can only observe the transmission process of the stress wave, and it's hard to obtain the actual dynamic information [29]. The theoretical analysis needs to adopt approximate and simplified methods, resulting in differences between the analysis results and actual results. Considering the great difficulty and economic cost of the experiment, it is very important to adopt numerical simulation. Based on the LS-DYNA® explicit dynamic analysis platform, the dynamic information of the detonation product can be obtained by using the numerical simulation method, and the crack propagation process can be visually displayed [30]. Scholars such as Yang, R.S. et al. [31] and Shen, T. et al. [32] have used numerical simulation to study the cutting seam cartridge, and the results show that the numerical simulation results are consistent with the measured data. However, most of the existing studies assume that the center of the blasthole and the cartridge are coincidental, that is, the concentric uncoupled cartridge structure. In actual engineering, the explosive will be close to one side of the blasthole due to gravity or misplacement, resulting in an eccentric uncoupled cartridge structure. When this structure explodes, the explosion energy does not uniformly act on the rocks around the blasthole and will produce local stress concentration effects around the blasthole wall. Especially for the cutting seam cartridge, due to the existence of cutting seam pipe, there is a large local concentrated load when the detonation products form the initial guided crack [33]. If the eccentric arrangement, the coupled side, and the uncoupled side will have a quiet blasting difference, which will affect the forming quality of the directional blasting and increase the damage to the surrounding rock [34]. At present, for the eccentric uncoupled structure of the cutting seam cartridge, the blasting dynamics and crack propagation law are still rarely studied. Therefore, it is necessary to study the eccentric uncoupled blasting effect of the cutting seam cartridge.

In this paper, based on the LS-DYNA® explicit dynamic analysis platform, a quasi-two-dimensional numerical model with five kinds of eccentric uncoupled coefficients is established by using the fluor-solid coupling algorithm. Based on the pressure cloud diagram and pressure-time interval curve output by the post-processing software, the dynamic response behavior characteristics of the explosive, air, the cutting seam pipe, and the rock mass in the eccentric and uncoupled case of the cutting seam cartridge are analyzed. And use the failure element method to simulate the crack propagation effect under completely eccentric conditions. Under the influence of different eccentric uncoupled coefficients, the distribution characteristics of the rock mass explosion stress field and the crack propagation law of the cutting seam cartridge blasting technology are studied to provide a theoretical basis for reducing the possible damage effects. It has important theoretical and practical significance to explore the role of stress waves in the vicinity of the blasting near district when the eccentric and uncoupled arrangement of the cutting seam cartridge and to deeply understand engineering phenomena, which can provide guidance and reference for future construction.

## 2. Methods

### 2.1. The Initial Crack Formation Mechanism of the Cutting Seam Cartridge

Since the pressure in the cutting seam direction is advanced and higher than the non-cutting seam direction, the cutting seam cartridge can control the formation of directional fracture. The formation of the initial crack in the cutting seam direction is mainly affected by two failure modes. The first one is the shear failure of radial shear stress caused by the compressive stress difference of rock mass under the action of the shock wave. The second is that because of the existence of the cutting seam pipe, the impact on the non-cutting seam direction is buffered so that the annular tensile stress is formed at the cutting seam direction.

The shear stress difference formed at the cutting seam of the blasthole wall:

$$t > S_{sd} \tag{1}$$

$$t = p_d - p_i \tag{2}$$

$$S_{sd} = \sigma tan\phi + C \tag{3}$$

where, $S_{sd}$ for dynamic shear strength of rock, $C$ for the rock dynamic cohesion, $\phi$ for rock dynamic friction angle.

When the maximum circumferential tensile stress on the blasthole wall is greater than the dynamic uniaxial tensile strength of the rock, the rock on the blasthole wall at the cutting seam will suffer tensile failure. Since the annular stress and radial stress have the following relationship:

$$\sigma = \mu p / (1 - \mu) \tag{4}$$

where, $\mu$ is poisson's ratio of the rock, $p$ is the pressure on the blasthole wall.

The conditions for the formation of initial cracks during the blasting of cutting seam cartridge obtained as follows:

$$p > (1 - \mu) \, S_{td} / \mu \tag{5}$$

$$p > (1 - \mu)(C - t) / (\mu \, tan\phi) \tag{6}$$

where, $S_{td}$ for dynamic uniaxial tensile strength of rock.

### 2.2. Model and Parameters

The eccentric and uncoupled numerical model of the cutting seam cartridge consists of four parts: explosive, cutting seam pipe, rock, and air. Among them, explosives and air use multi-material *ALE* algorithm to avoid solving anomalies caused by large deformations, and *Lagrange* elements are used for rocks and cutting seam pipe. A penalty function-based fluid-solid coupling constraint is set between the fluid substance and the *Lagrange*

element to transfer the blasting load to the cutting seam pipe and the rock. The *CON-TACT_AUTOMATIC_SURFACE_TO_SURFACE* contact is set between the cutting seam pipe and the blasthole wall to realize energy transfer. To reduce the amount of numerical calculation, an element length is adopted in the thickness direction, and a displacement constraint is imposed. Since the end effect of the explosive volume is not considered, the three-dimensional problem can be converted into a two-dimensional problem for research, that is, a quasi-two-dimensional calculation model is used for simulation. The finite element model is shown in Figure 1. The model has a total of 623,536 elements. The model measuring is 600 mm × 600 mm × 1 mm. The blasthole diameter is 40 mm, the explosive diameter is 24 mm, and the uncoupled coefficient between the explosive and the blasthole is 1.67 [35]. The inner wall of the cutting seam pipe is close to the explosive, the thickness is 4 mm, and the cutting seam width is 4 mm [36–38]. Because the experimental model is small, in order to simulate the actual infinite rock mass, transmission conditions are imposed on the model boundary, and the boundary is set as a non-reflective boundary.

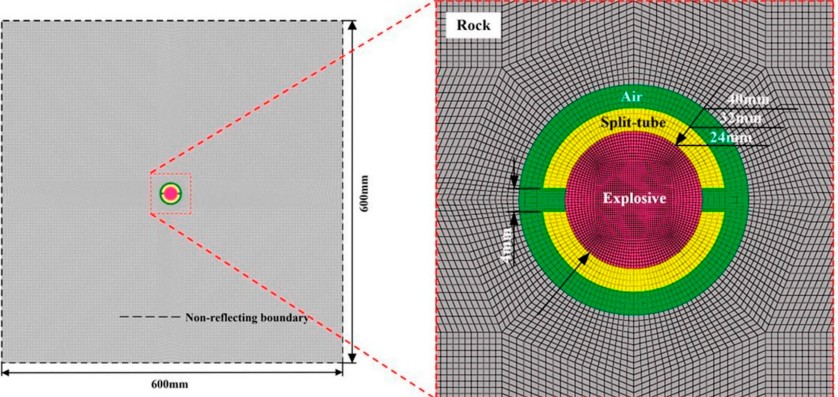

**Figure 1.** Geometric model parameters and partial enlarged grid.

### 2.3. Working Conditions Establishment

To describe the eccentricity of the cartridge in the blast hole, the inside of the blasthole can be divided into the coupled side and the uncoupled side, as shown in Figure 2. The degree of deviation of the cutting seam cartridge from the center is defined as the eccentric uncoupled coefficient [39]. Since the thickness of the cutting seam pipe determines the eccentric uncoupled coefficient, the expression is:

$$E = \frac{OO_2}{R_1 - R_2} \tag{7}$$

where $E$ is the eccentric uncoupled coefficient, $R_1$ is the radius of the blasthole, $R_2$ is the radius of the cutting seam pipe, and $OO_2$ is the distance from the center of the cartridge to the center of the blasthole. When the eccentric uncoupled coefficient is 0, the cartridge coincides with the center of the blasthole. When the eccentric uncoupled coefficient is 1, the coupled side cutting seam pipe is close to the blasthole wall.

In order to study the blasting effect of the eccentric uncoupled structure of the cutting seam cartridge, five groups of working conditions with eccentric uncoupled coefficients of 0.0, 0.25, 0.5, 0.75, and 1.0 were set up for research. The working conditions are shown in Table 1.

**Table 1.** Working condition setting and corresponding eccentric uncoupled coefficient.

| Eccentric Uncoupled Coefficient | 0.0 | 0.25 | 0.5 | 0.75 | 1.0 |
|---|---|---|---|---|---|
| Working condition | $E_0$ | $E_1$ | $E_2$ | $E_3$ | $E_4$ |

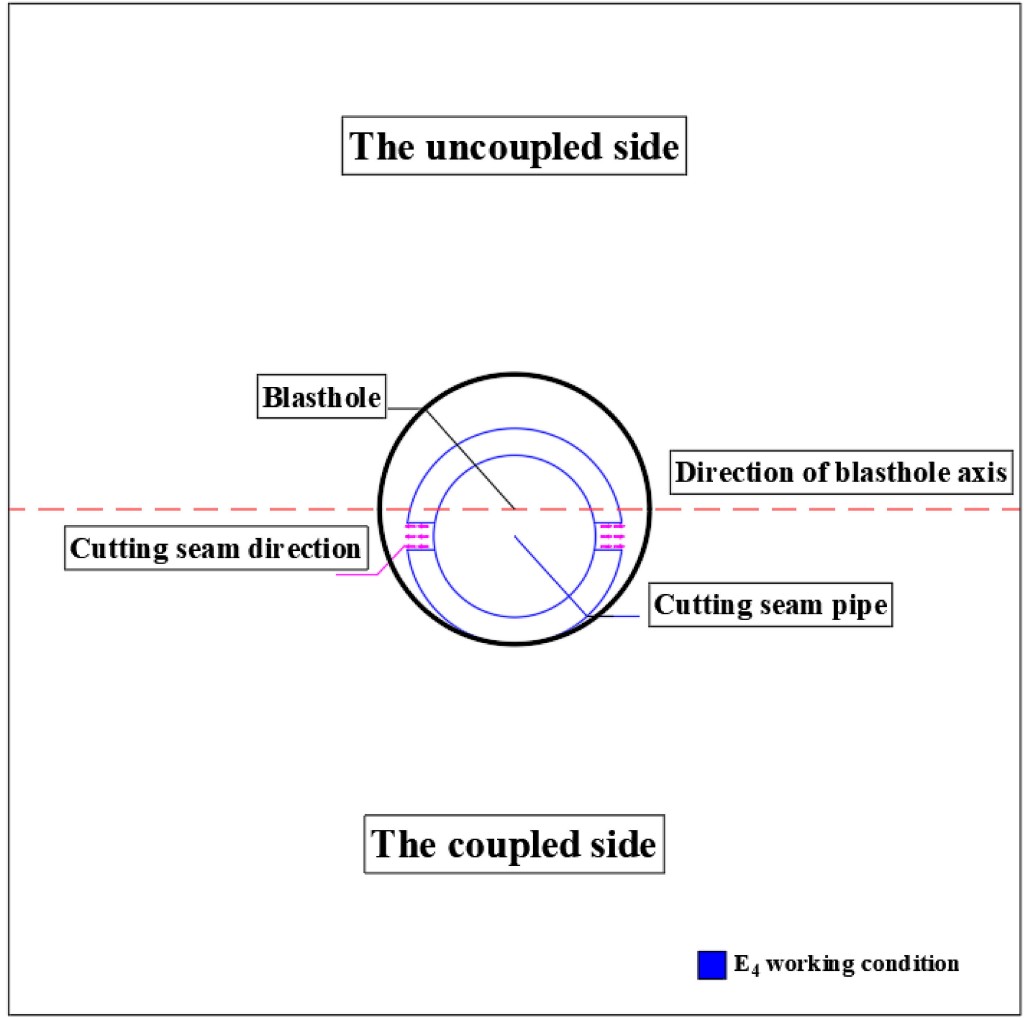

**Figure 2.** Schematic diagram of model partition.

*2.4. Material Parameters*

2.4.1. Model Unit System

Since the finite element software only performs numerical calculations, and its units are assigned from outside, we need to uniformly regulate the unit system. The prescribed results are shown in Table 2.

**Table 2.** Unit system of the model.

| Length | Force | Time | Velocity | Mass | Stress | Density |
|--------|-------|------|----------|------|--------|---------|
| cm | N | μs | cm/μs | g | Mbar = $10^{11}$ Pa | g/cm$^3$ |

After the unit system is determined, when inputting the parameters, it needs to be converted according to the unit system.

2.4.2. Explosive Material Parameters

The explosive material is 2$^\#$ rock emulsion explosive. The explosive material parameters are shown in Table 3.

**Table 3.** Explosive material parameters.

| $\rho/(\text{kg·m}^{-3})$ | $D/(\text{m s}^{-1})$ | CJ/Pa | A/GPa | B/GPa | $R_1$ | $R_2$ | $\omega$ | E/GPa |
|-----------|-----------|-------|-------|-------|-------|-------|------|-------|
| 1.1 | 0.36 | 3.5 | 214.4 | 0.182 | 4.2 | 0.9 | 0.15 | 4.192 |

In the numerical model, the keyword *MAT_HIGH_EXPLOSIVE_NURN* and the *EOS_JWL* are used to describe the material model of explosives. The state equation *JWL* replaces the load-time history curve by calculating the pressure in the element after the explosive is detonated, which can better describe the relationship between the pressure and the specific volume in the explosive process.

The state equation of explosive is as follows:

$$P = A\left(1 - \frac{\omega}{R_1 V}\right)e^{-R_1 V} + B\left(1 - \frac{\omega}{R_2 V}\right)e^{-R_2 V} + \frac{\omega E}{V} \tag{8}$$

where, $A$, $B$, $R_1$, $R_2$, $\omega$ are the material constants, $P$ is the pressure, $V$ is the relative volume, $E_0$ is the initial specific internal energy.

### 2.4.3. Air Material Parameters

The air medium adopts the *MAT_NULL* blank material model and is described according to the *EOS_LINEAR_POLYNOMIAL* linear polynomial equation of state, and the relative volume $V_0 = 1.0$. The state equation is shown in Equation (9), and the parameters are shown in Table 4.

$$P = C_0 + C_1\mu + C_2\mu^2 + C_3\mu^3 + \left(C_4 + C_5\mu + C_6\mu^2\right)E \tag{9}$$

where $C_0, C_1, C_2, C_3, C_4, C_5, C_6$ are input parameters; E is the internal energy parameter;

**Table 4.** Air materials and state equation parameters.

| $\rho/(kg\cdot m^{-3})$ | $C_0$ | $C_1$ | $C_2$ | $C_3$ | $C_4$ | $C_5$ | $C_6$ |
|---|---|---|---|---|---|---|---|
| 1.29 | $-1 \times 10^5$ | 0 | 0 | 0 | 0.4 | 0.4 | 0 |

### 2.4.4. Rock Material Parameters

Under the action of blasting load, the rock mass will be subjected to a high strain rate. Therefore, the plastic kinematic hardening model *MAT_ PLASTIC_KINNEMATIC*, which takes into account the influence of strain rate on the strength of rock mass, is selected, and the Cowper-Symonds parameters C and P in the model are also considered. The rock mass parameters are shown in Table 5 [40].

**Table 5.** Rock material parameters.

| $\rho/(g\cdot cm^{-3})$ | E/GPa | $\mu$ | $\sigma_s$/MPa | $E_{max}$/GPa | $\sigma_c$/MPa | $\sigma_t$/MPa | $\beta$ | $C/s^{-1}$ | P |
|---|---|---|---|---|---|---|---|---|---|
| 2.7 | 68.69 | 0.228 | 75 | 40 | 150 | 5.6 | 0.6 | 2.63 | 3.96 |

The explosion will release a large amount of energy in a short time, forming a high temperature and pressure environment and exposing the rock mass to a high strain rate. The dynamic compressive strength of rock increases with the increase of the loading strain rate. At this time, the dynamic tensile strength and the static tensile strength are in a power function relationship [41]. In engineering blasting, the rock loading strain rate $\varepsilon$ is generally $10^0 \sim 10^5$ s$^{-1}$. Under the combined action of explosive gas and stress waves, the failure of the rock mass can be divided into compression failure in the crushing zone and tensile failure in the fracture zone. In the crushing zone, the strain rate is $10^4$ s$^{-1}$; in the fracture zone, the strain rate is $10^2$ s$^{-1}$.

Therefore, the Von Mises failure criterion is adopted to obtain the rock mass failure criterion:

$$\sigma_0 = \sigma_d \varepsilon_d^{1/2} \tag{10}$$

where $\sigma_0$ is the uniaxial failure strength of the rock, when the Von Mises stress in the rock mass exceeds this value, the rock mass is considered to be broken and failed; $\sigma_d$, $\varepsilon_d$ are the static strength and strain rate of the crushed zone or the fracture zone.

### 2.4.5. Material Parameters of Cutting Seam Pipe

The cutting seam pipe is similar to a thin-walled shell structure. Due to the existence of cutting seam pipe, the equilibrium of detonation products on the blasthole is changed, and the energy is further accumulated in the cutting seam direction, thus strengthening the damage to the blasthole wall in the cutting seam direction [42]. Under the strong impact load of the explosion, the cutting seam pipe will eventually deform or even be damaged. Since the cutting seam pipe is subjected to a greater strain rate during blasting ($>10^5$ s$^{-1}$), the high-conductivity oxygen-free copper pipe material is selected, and *MAT_STEINBERG* is used as the model to simulate the cutting seam pipe material. The model only defines the state of the material before it melts, and the model expression is:

$$G = G_0 \left[ 1 + \left( \frac{G'_p}{G_0} \right) \frac{P}{\eta^{1/2}} + \left( \frac{G'_p}{G_0} \right)(T - 300) \right] \tag{11}$$

$$\sigma_y = \sigma_0 [1 + \beta(\varepsilon + \varepsilon_0)]^n \left[ 1 + \frac{\sigma'_p}{\sigma_0} \frac{p}{\eta^{1/3}} + \frac{G'_T}{G_0}(T - 300) \right] \tag{12}$$

where $\eta = V_0/V$ is the compression ratio, $G'_p$, $\sigma'_p$, $G'_T$ respectively represent the partial derivatives of $G$ and $\sigma$ with respect to pressure $p$ and temperature $T$, $\beta$ and n are work hardening parameters, $\varepsilon$ is equivalent plastic strain.

At the same time, the solid high-pressure *Grüneisen* equation of state is used for description. The state equations for compressed and stretched materials are shown in Equation 13, and the parameters are given in Table 6.

$$p = \frac{\rho_0 C^2 \mu \left[ 1 + \left( 1 - \frac{\gamma_0}{2} \right) \mu - \frac{a}{2} \mu^2 \right]}{\left[ 1 - (S_1 - 1)\mu - S_2 \frac{\mu^2}{\mu+1} - S_3 \frac{\mu^3}{(\mu+1)^2} \right]^2} + (\gamma_0 + a\mu)E \tag{13}$$

$$p = \rho_0 C^2 \mu + (\gamma_0 + a\mu)E \tag{14}$$

where $C$ is the intercept of the $u_s - u_p$ curve, $S_1$, $S_2$, $S_3$ are the slope coefficients of the $u_s - u_p$ curve, $\gamma_0$ is Grüneisen coefficient, a is the first-order correction coefficient for $\gamma_0$, $\mu$ is the density parameter, $\mu = \frac{\rho}{\rho_0} - 1$.

**Table 6.** Material parameters of cutting seam pipe.

| $\rho/(\text{g·cm}^{-3})$ | $G_0/GPa$ | $\sigma_0/GPa$ | $\beta$ | n | $\gamma_1$ | $\sigma_m/GPa$ | b | $b'$ | h |
|---|---|---|---|---|---|---|---|---|---|
| 8.93 | 47.7 | 0.12 | 36.0 | 0.45 | 0 | 0.64 | 2.83 | 2.83 | $3.77 \times 10^{-4}$ |

| f | A | $T_{mo}/K$ | $\gamma_0$ | a | $p_{cut}/GPa$ | $K_{spall}$ | $C/(\text{m s}^{-1})$ | $S_1$ | $A_*$ |
|---|---|---|---|---|---|---|---|---|---|
| 0.001. | 63.5 | 1790.0 | 2.02 | 1.4 | −9.0 | 3.0 | 3940 | 1.49 | 0.47 |

Note: $A_*$ is A in the Grüneisen equation of state.

## 3. Simulation Results and Analysis

### 3.1. Explosion Stress Field Simulation Results and Analysis

To observe the dynamic propagation process of the stress wave and explosive gas in the stress field of the rock mass medium, when the cutting seam cartridge exploded. The stress field of the rock under the working condition is simulated. The pressure cloud diagrams at different times are shown in Figure 3.

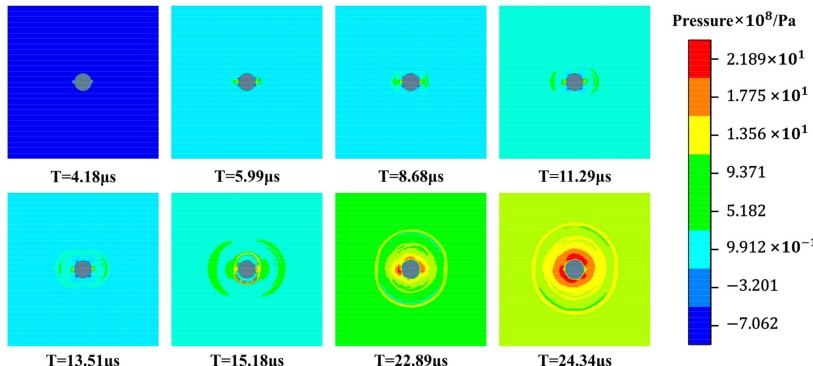

**Figure 3.** Pressure cloud diagram at different moments under $E_0$ working condition.

When t = 5.99 μs, the initial detonation product rushes out of the cutting seam pipe, and the detonation wave contacts the air at the cutting seam to form an initial shock wave. The initial shock wave rushes out along the cutting seam direction and acts on the blasthole wall. When t = 8.68 μs, the existence of a cutting seam pipe makes the detonation products preferentially act on specific positions. A small part of the explosive gas flows around the interface and is used at the non-cutting seam of the blasthole wall. It can be seen from the figure that there is a sudden change in pressure that moves up from the cutting seam along the circumference of the blasthole, and the shape is similar to an "X" shape. When the shock wave acts on the wall of the blasthole and the cutting seam pipe wall, part of the reflected wave and transmission are generated. When t = 15.18 μs, the two circumfluence products converge at the vertical cutting seam direction, which strengthens the energy of the rock mass and forms the initial pressure peak to the vertical cutting seam direction.

At t = 22.89 μs, the cutting seam pipe expands continuously and comes into contact with the blasthole wall, and the shock wave spreads outward in the form of a circular surface wave. When t = 24.34 μs, the cutting seam pipe loses its wrapping property to the detonation product. It's known from the pressure cloud diagram that the detonation product is released gradually with the expansion of the cutting seam pipe when the cutting seam cartridge explodes. Figure 4 is a timing diagram of the cutting seam pipe being expanded and gradually destroyed by the impact of detonation products. Due to the existence of the cutting seam pipe, the stress is concentrated at the cutting seam direction, which enhances the guiding effect of the initial crack and cushions the stress at the non-cutting seam direction. Therefore, in directional blasting, the cutting seam between adjacent blast holes are arranged along the axis of the blast hole to achieve the effect of directional blasting.

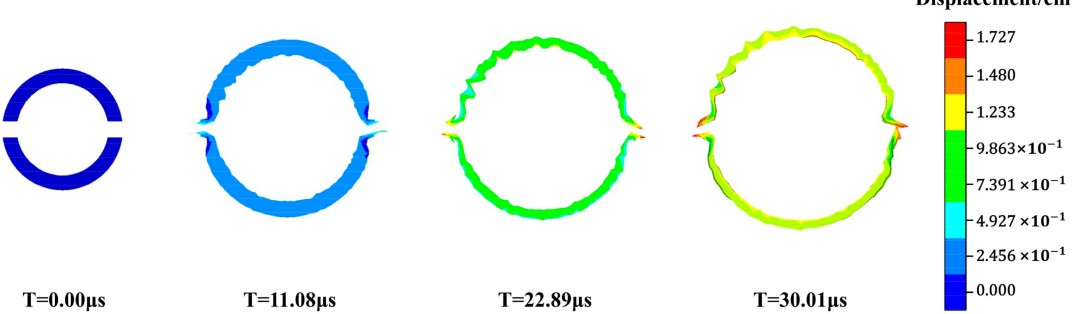

**Figure 4.** Dynamic deformation diagram of cutting seam pipe.

### 3.2. Analysis of Explosion Pressure Time and Space Distribution

#### 3.2.1. Direction of Blasthole Axis

As the density of cutting seam pipe is much higher than detonation products, the detonation products are preferred to be applied to the cutting seam direction of the blasthole wall to form the initial guiding crack, so as to form a good directional blasting effect. In order to study the temporal and spatial distribution of the explosion pressure at the axis of the blasthole. Along the axis of the blasthole, one measuring point is selected from the blasthole wall every 10mm, and a total of 10 measuring points are selected. The arrangement of measuring points is shown in Figure 5.

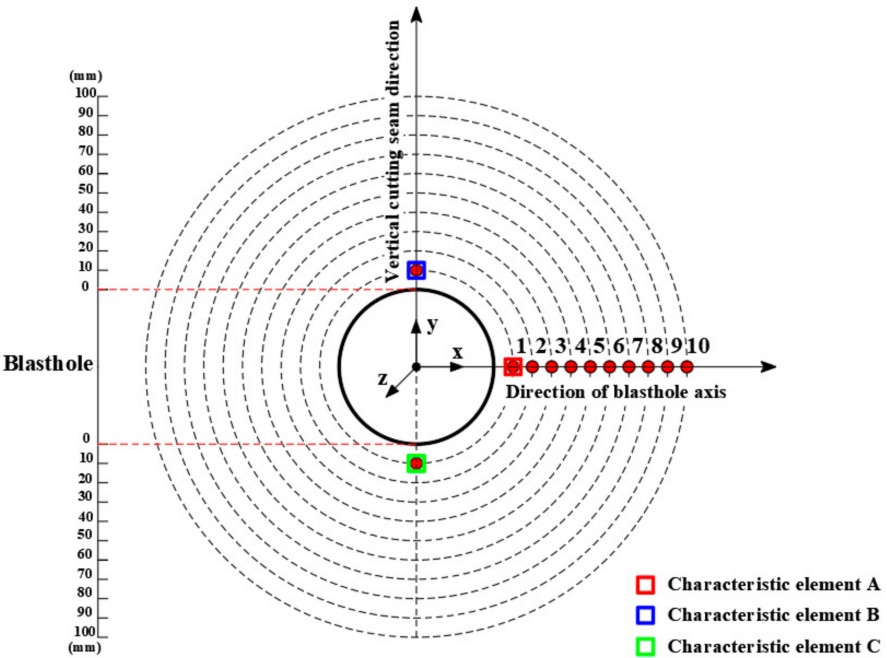

**Figure 5.** The arrangement of measuring points and characteristic element.

The effect of detonation products on the rock mass is very complicated. It is not strictly the dynamic effect of the explosive stress wave first, and then the quasi-static effect of the explosive gas pressure. Figure 6 shows the pressure-time interval curve of the measuring point under $E_0$ working condition. According to the pressure-time interval curve, the whole process is divided into 3 stages, namely the initial stage, the fluctuation stage, and the peak stage. During the explosion of explosives, the speed of shock wave propagation is higher than that of explosive gas [43]. In the initial stage, the detonation wave impacts the cutting seam pipe. The initial shock wave is formed at the cutting seam and rushes out along the cutting seam direction, and the initial shock wave disturbs the rock mass unit to a peak. In the fluctuation stage, the explosive gas, shock wave, and reflected wave act together. At this time, the wrapping property of the cutting seam pipe still exists. The existence of the reflected wave makes the rock mass to be repeatedly impacted, showing the effect of fluctuation in the pressure-time history curve. In the peak stage, the cutting seam pipe continuously expands and contacts the blasthole wall. The cutting seam pipe gradually loses its wrapping property to the detonation products. The detonation products are released comprehensively, transferring a large amount of energy to the surrounding rock mass, and the rock mass pressure gradually reaches the peak value.

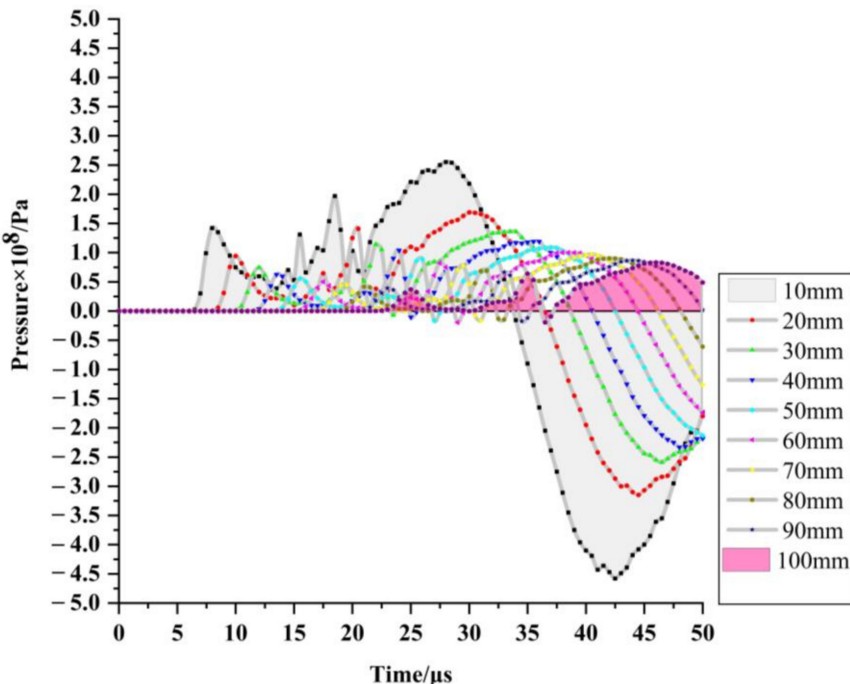

**Figure 6.** Pressure-time interval curve of axial measuring point.

The decay rate of detonation pressure decreases from fast to slow during the propagation of rock mass. Figure 7 extracted the initial peak pressure of 10 measuring points at the blasthole axis under $E_0$ working condition. The initial shock wave decreases in the form of power function when it is transferred in the rock mass, which is following the law of pressure attenuation in the blasting of cutting seam cartridge [44].

$$y = a\,x^b \tag{15}$$

where $y$ is the initial pressure peak, $a$ and $b$ are the fitting coefficients, and $b$ represents the rate of attenuation.

As the propagation distance of the shock wave increases, the shock wave continues to attenuate and finally attenuates into a stress wave. The attenuation rate of a shock wave at 8~10 μs was 6.9 times that at 22~26 μs, and the peak pressure at measuring point 1 was 2.83 times that at measuring point 10. The reason why this phenomenon exists is that the existence of the cutting seam pipe guides the initial crack at the cutting seam direction. It is mainly manifested in the rock mass near the area and the crack propagation in the far area is extended by the quasi-static gas action. Since the shock wave decays faster with the increase of distance, the follow-up study only discusses the dynamic shock process in the near rock mass.

The penetration and flatness of the blasthole axis control the effect of directional blasting. When the cartridge is arranged eccentrically, the load in the blasthole is unevenly distributed. Rock mass characteristic element A is selected to study the pressure change with time under the conditions of $E_0 \sim E_4$, as shown in Figure 8. With the gradual increase of the eccentric uncoupled coefficient, the cutting seam gradually moves away from the blasthole axis and the stress at each stage of element A reduced.

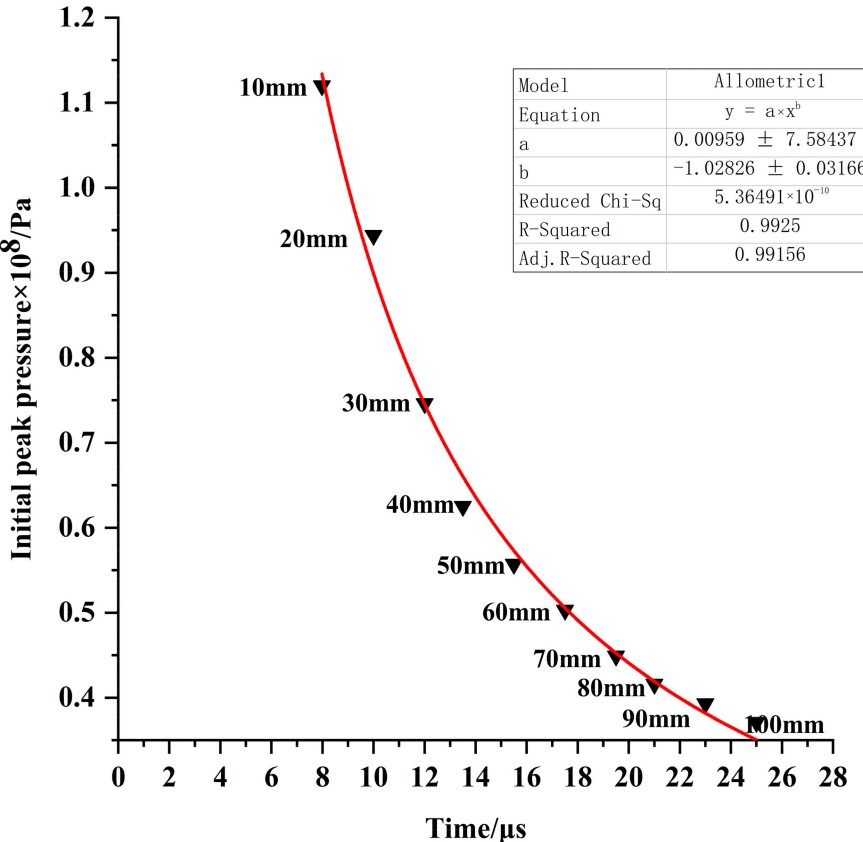

| Model | Allometric1 |
| --- | --- |
| Equation | $y = a \times x^b$ |
| a | $0.00959 \pm 7.58437$ |
| b | $-1.02826 \pm 0.03166$ |
| Reduced Chi-Sq | $5.36491 \times 10^{-10}$ |
| R-Squared | 0.9925 |
| Adj.R-Squared | 0.99156 |

**Figure 7.** Pressure-peak curve of axial measuring point.

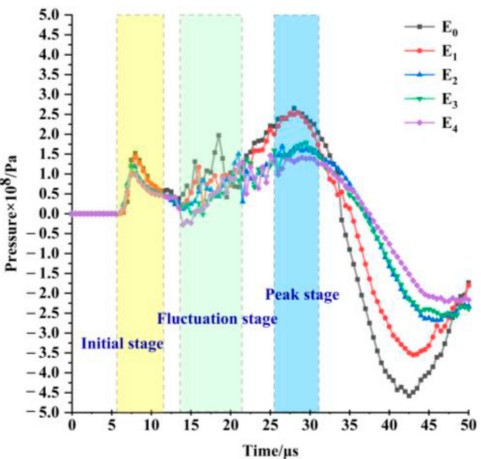

**Figure 8.** Pressure-time interval curve of A measuring point under different working conditions.

Figure 9 extracts the pressure-time interval curve of $E_0$ and $E_4$ working conditions. Due to the good energy gathering of the cutting seam pipe, when the cutting seam is deflected from the axis to the coupled side of the blasthole, the stress concentration effect at the blasthole axis is weakened. Therefore, when the cutting seam cartridge is eccentric and uncoupled, the cracking effect of the detonation product at the blasthole axis is weakened, which will affect the directional blasting effect.

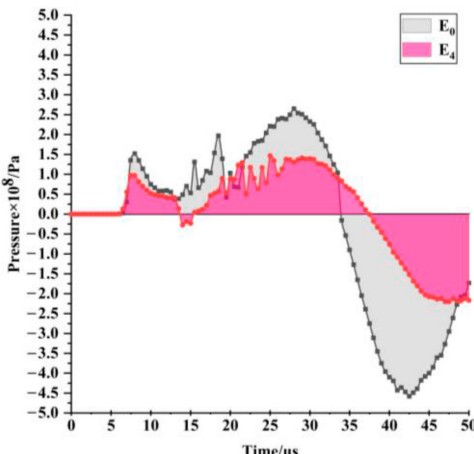

**Figure 9.** Pressure-time interval curve of $E_0$ and $E_4$ working conditions.

With the increase of the eccentric uncoupled coefficient, the action time of stress wave and detonation gas is delayed, and the pressure peaks at each stage gradually decrease, as shown in Figure 10. From $E_0$ to $E_4$ working conditions, the peak time of the fluctuation stage is delayed by 3.5 μs, and the peak pressure of $E_4$ is reduced by 52.7% compared with $E_0$. The peak time of the peak stage is delayed by 1.5 μs, and the peak pressure of $E_4$ is reduced by 92.1% compared with $E_0$. The peak pressure is determined by the wrapping property of the cutting seam pipe and the eccentric uncoupled coefficient in the fluctuating stage. The initial detonation products mainly rush out along the cutting seam direction, and a small part of the detonation products flow around and reflect along the cutting seam pipe. The eccentric uncoupled structure results in a larger airfield on the uncoupled side and the wrapping property of the cutting seam pipe leads to a large stress attenuation on the blast hole axis. In the peak stage, the cutting seam pipe deforms seriously and gradually loses its wrapping properties, and the peak pressure is determined by the eccentric uncoupled coefficient. The air area on the uncoupled side consumes part of the detonation energy, and the larger air area delays the time for the cutting seam pipe to contact the blasthole wall. Therefore, the eccentric uncoupled coefficient affects the blasting effect at the blasthole axis. A larger eccentricity will weaken the formation of initial cracks, leading to over-excavation or under-excavation.

### 3.2.2. Vertical Cutting Seam Direction

When traditional explosives are arranged eccentrically, the damage range of the coupled side increases, and the fracture zone decreases. The fracture zone on the uncoupled side increases, the damage range decreases, and the blasthole pressure is not evenly distributed. Due to the existence of the cutting seam pipe, the shock wave at the non-cutting seam direction can be buffered, but the force of the rock mass at the vertical cutting seam directly under the eccentric arrangement is not known. In order to study the impact dynamics behavior of rock mass elements on the coupled side and uncoupled side, a rock mass characteristic elements B and C, which are 10 mm away from the blasthole wall, are selected in the vertical cutting seam direction, and the arrangement of measuring points is shown in Figure 5.

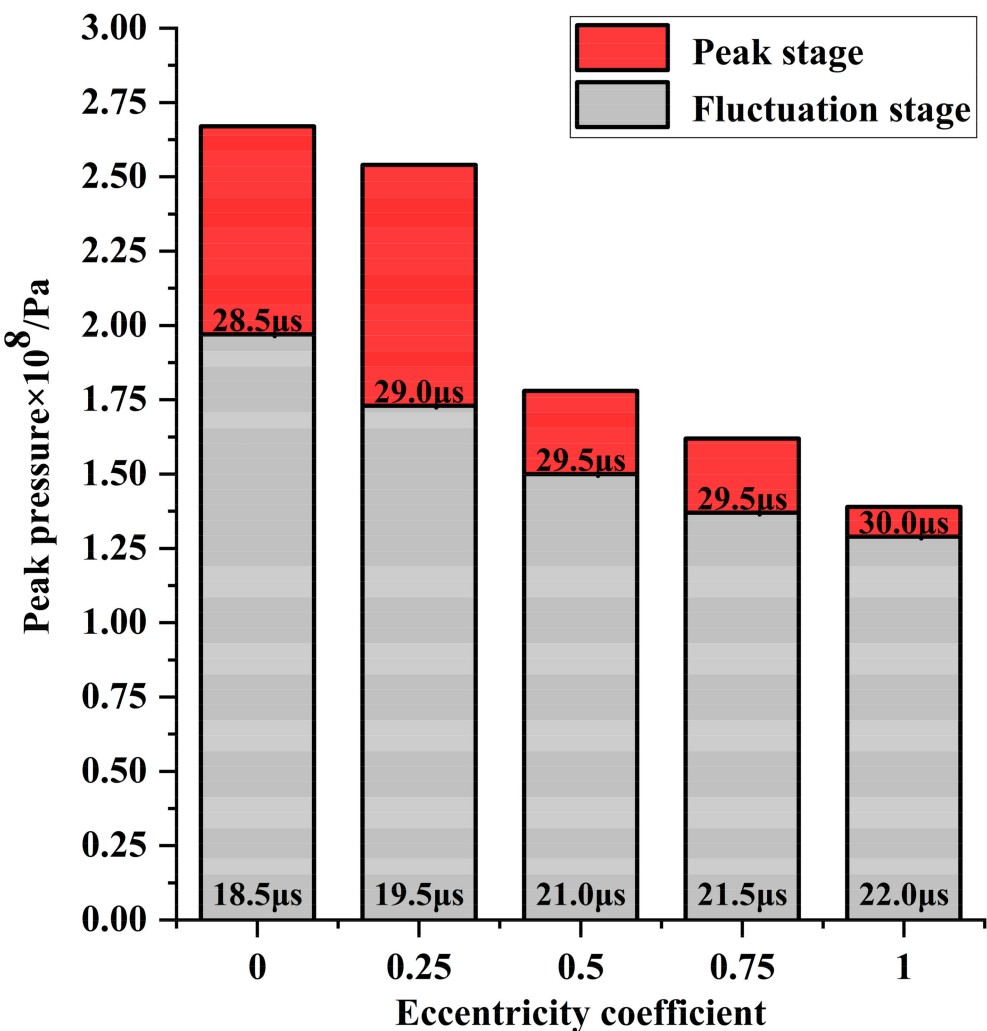

**Figure 10.** Peak pressure diagram of A measuring point under different working conditions.

The pressure-time interval curve at the characteristic element B on the coupled side is shown in Figure 11. There are still three stages in the process of impact: initial stage, fluctuation stage, and peak stage. In the initial stage, due to the existence of the cutting seam pipe, a large number of shock waves impact the blasthole and the cutting seam pipe to form a reflected wave, and only a small amount is transmitted through the cutting seam pipe. In the fluctuation stage, the explosive gas flows around the interface between the cutting seam pipe and the blasthole, which converges in the vertical cutting seam direction. With the increase of the eccentric uncoupled coefficient, the air area on the uncoupled side increases, the detonation energy consumption increases during the transmission process. The impact in the vertical cutting seam direction is weakened, and the sudden period of the pressure fluctuation stage is shortened. With the severe expansion and deformation of the cutting seam pipe, the rock mass pressure gradually reaches its peak. The moments of each stage about the shock wave are delayed, and the peak pressure is greatly reduced, as shown in Figure 12.

It can be seen from Figure 13 that the peak stage of $E_4$ is 14.0 μs later than the fluctuation stage, and $E_0$ is only delayed by 7.5 μs. This is because the cutting seam pipe deviates to the coupled side, and the time it takes for the cutting seam pipe to expand and contact the blasthole wall increases, resulting in the peak time to be delayed. When the eccentric uncoupled coefficient is greater than 0.5, the pressure weakens significantly, and the pressure peak of the $E_0$ at the peak stage is 6.76 times that of the $E_4$. It shows that when

the eccentric uncoupled coefficient increases, the buffer effect on the uncoupled side of the cutting seam cartridge is further enhanced, and the impact of blasthole is reduced.

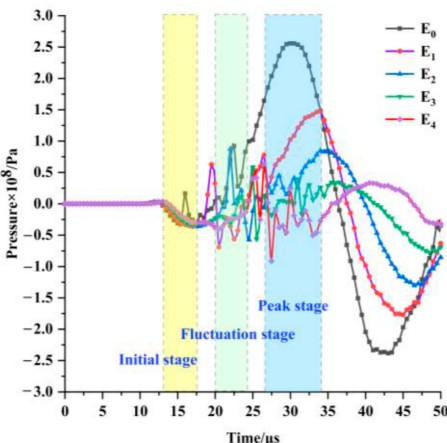

**Figure 11.** Pressure-time interval curve of B measuring point under different working conditions.

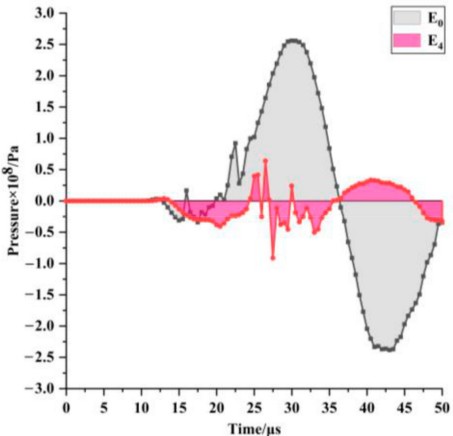

**Figure 12.** Pressure-time interval curve of $E_0$ and $E_4$ working conditions.

The pressure-time interval curve at characteristic element C is shown in Figure 14. As the eccentric uncoupled coefficient increases, the distance between the cutting seam pipe on the coupled side and the blasthole decreases. After expansion, the cutting seam pipe contacts the blasthole more quickly and releases detonation energy. The air area on the coupled side is reduced so that the detonation products gather in the air area of a small space.

As shown in Figure 15, when the eccentric uncoupled coefficient is 1.0, that is when the bottom of the cutting seam cartridge is in full contact with the blasthole, the coupled side is subjected to greater pressure impact in the initial stage, and the damage to the rock mass is increased.

It can be seen from Figure 16 that $E_4$ and $E_3$ reach the peak pressure in the initial stage of 8~9 μs. Compared with $E_2$, the pressure peak value of $E_4$ is increased by 1.4 times, and the peak time is advanced by 20.5 μs. This is because the smaller air spacing on the coupled side accelerates the time of the shock wave acting on the blasthole, which causes the initial detonation products to gather in a small space to form the stress concentration effect.

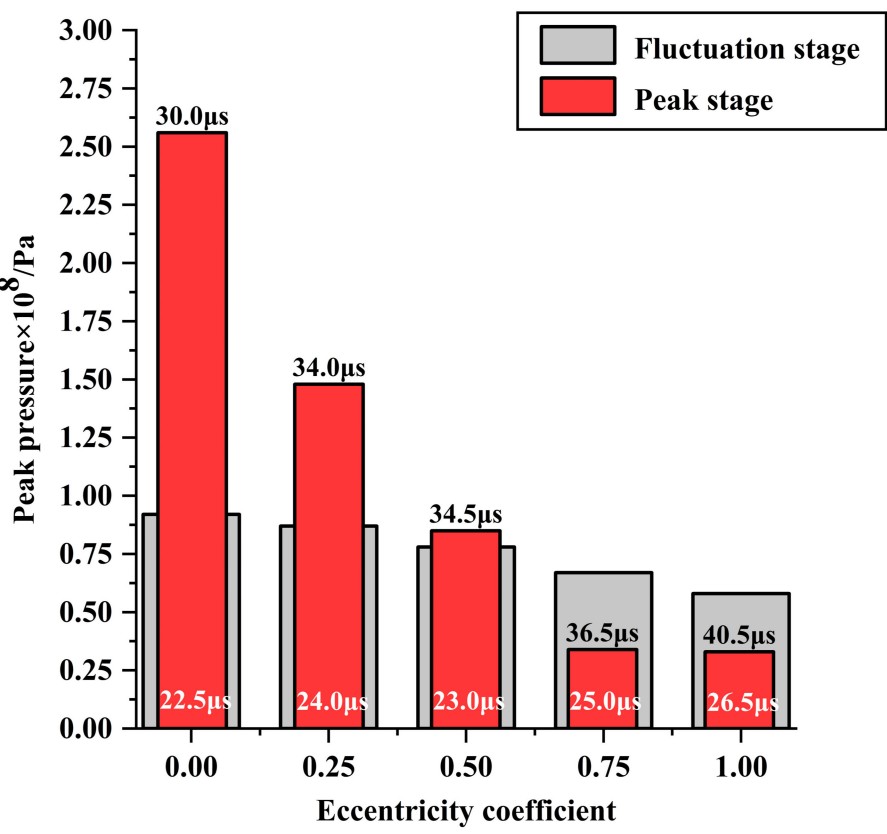

**Figure 13.** Peak pressure diagram of B measuring point under different working conditions.

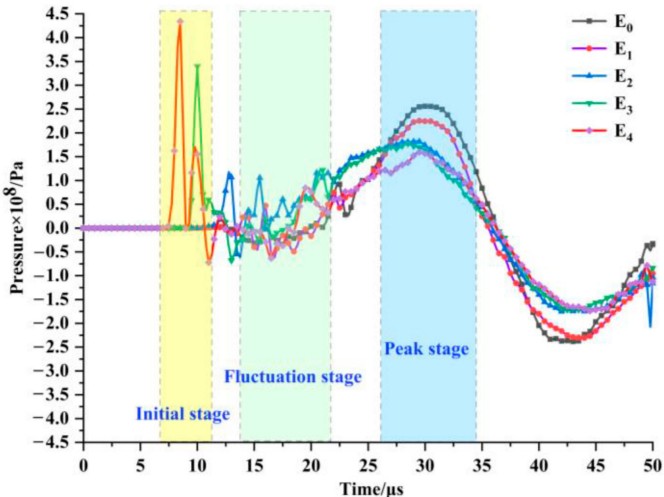

**Figure 14.** Pressure-time interval curve of C measuring point under different working conditions.

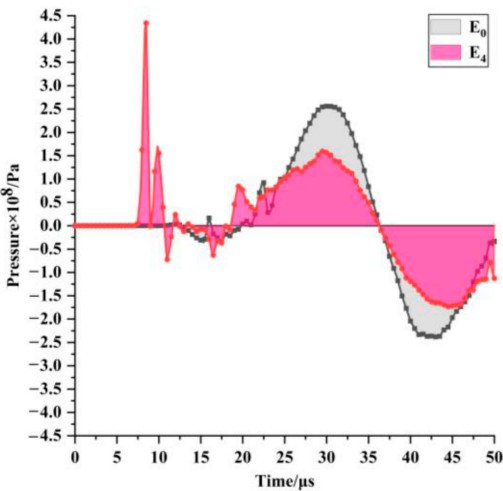

**Figure 15.** Pressure-time interval curve of $E_0$ and $E_4$ working conditions.

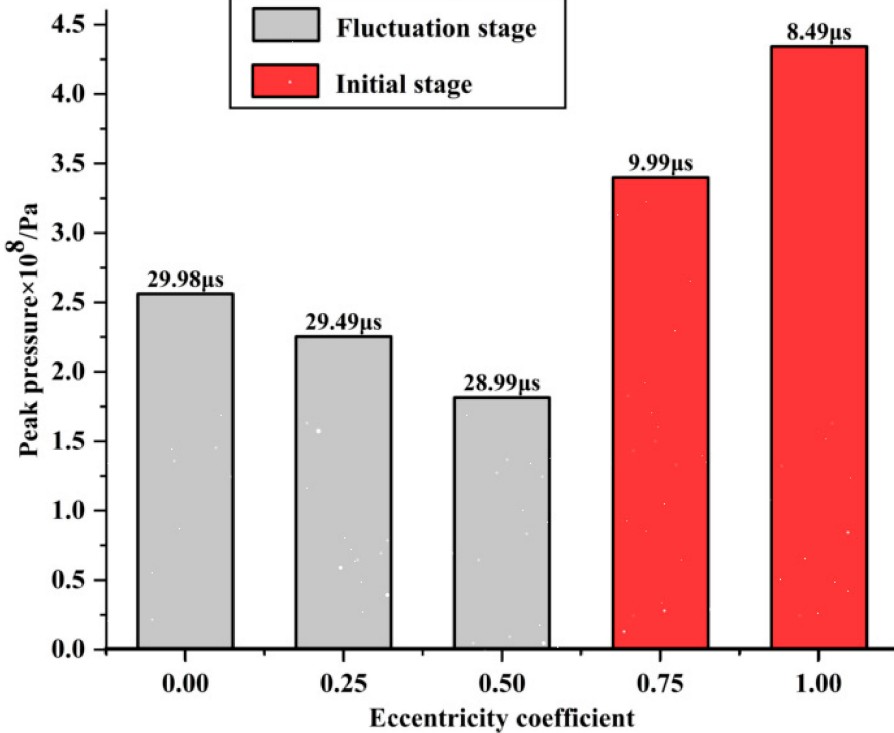

**Figure 16.** Peak pressure diagram of C measuring point under different working conditions.

Figure 17 compares the peak pressures of characteristic element C on the coupled side and characteristic element B on the uncoupled side. The $E_0$ belongs to the concentric uncoupled structure, with the same width of air interval between the cartridge and the blasthole, so the stress condition is the same. With the increase of the eccentric uncoupled coefficient, the pressure difference between the coupled side and the uncoupled side gradually increases. The peak pressure of the coupled side under $E_4$ working condition is 5.78 times that of the uncoupled side. It indicates the eccentric and uncoupled arrangement of the cutting seam cartridge, the load is unevenly distributed, and the rock mass damage on the coupled side and the uncoupled side is quite different, which requires attention in actual construction. If the coupled side is a rock that needs to be retained, the effect of directional blasting may be poor.

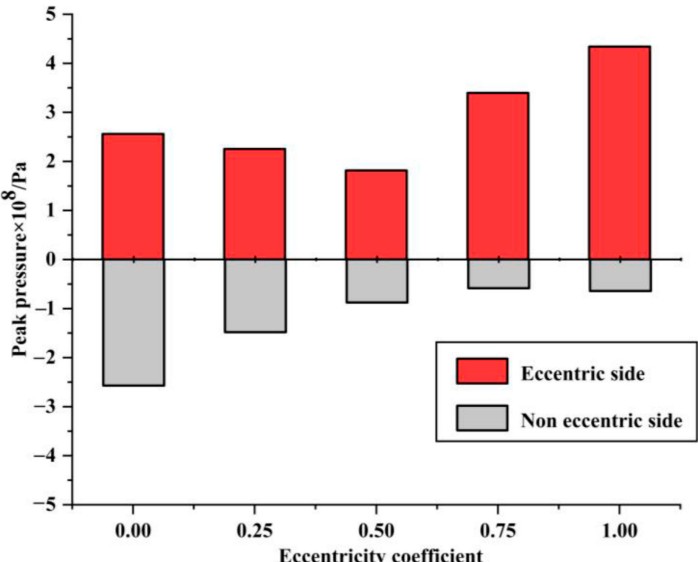

**Figure 17.** Comparison of peak pressure at measuring points B and C.

*3.3. Circumferential Air Domain Pressure Analysis*

When the cutting seam cartridge explodes, explosive gas and detonation waves rush out along the cutting seam pipe, and initial cracks are formed in the blasthole wall at the cutting seam, which continues to expand under the action of subsequent explosive gas. Part of the detonation products flows around the cutting seam pipe and acts on other parts of the blasthole wall. In order to study the effect of detonation products on the initial crack under the condition of the eccentric and uncoupled arrangement, and to verify the correctness of the above conclusions. Circumferentially select the air element at 4 mm outside the cutting seam pipe. The measuring points are arranged as shown in Figure 18, and the cutting seam direction is 90° and 270°.

Figure 19 shows the initial pressure peak in the form of a bar graph, showing the distribution of the initial pressure on the air characteristic element under $E_0 \sim E_4$ working conditions. The stress field generated by the detonation product after the explosion is symmetrical about the vertical cutting seam direction. In $E_0$ and $E_1$ working conditions, the eccentric uncoupled coefficient is small, and a large number of detonation products are punched out vertically along the cutting seam direction (90°, 270°), impacting the rock mass element to form the initial cracks. The initial pressure vertical to the cutting seam direction (0°, 180°) is second only to the cutting seam direction, which is formed by the convergence and superposition of the left and right circumfluence gas and part of the shock wave. With the increase of the eccentric uncoupled coefficient, the detonation products punched out of the cutting seam direction in the range of 90°~120° and 240°~270°, forming a larger stress concentration on the coupled side.

It can be seen from the working conditions $E_2$, $E_3$ and $E_4$ that the 180° of the coupled side has received a greater impact. This is because the air area on the coupled side is reduced, the circumfluence gas and the shock wave are weakened by the impedance of the air, and the expanded of cutting seam pipe can contact the blasthole wall and transfer energy in a short time. There are sudden changes in pressure values in the ranges of 120°~150° and 210°~240°. This is due to the circumfluence of detonation products along the cutting seam pipe, which is consistent with the observation in Figure 3. Because the density of the cutting seam pipe is much greater than that of the air medium, the explosion product separates from the interface when it flows around, resulting in the local drop of pressure value being formed. Observing the bar graph of $E_0 \sim E_4$ working conditions, it can be seen that as the eccentric uncoupled coefficient increases, the load distribution in the blasthole has a large deviation to the coupled side. A greater stress concentration effect will

be formed at the cutting seam and the coupled side, which is consistent with the results of the previous analysis.

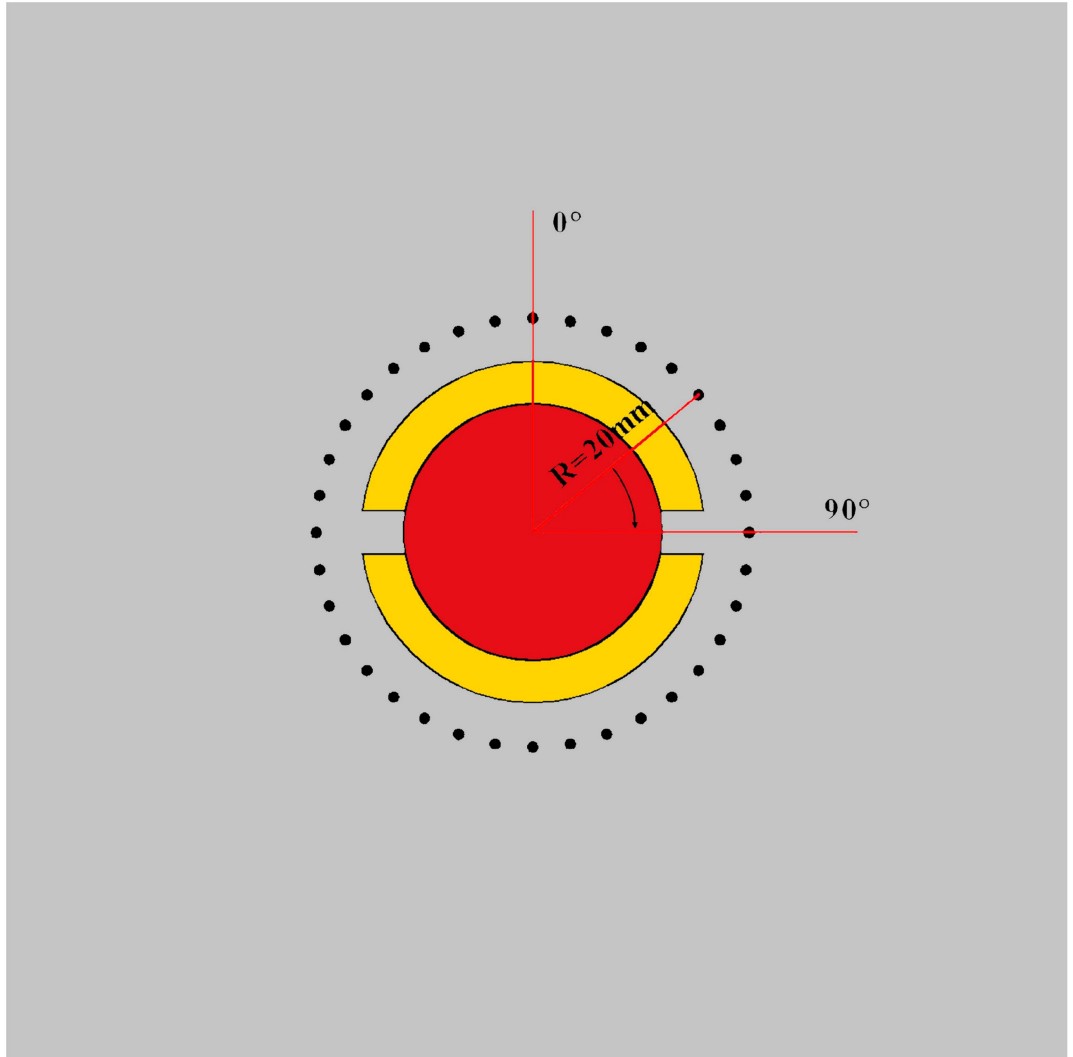

**Figure 18.** The arrangement of circumferential air measuring points.

### 3.4. Analysis of Rock Damage around the Blasthole

In order to simulate the crack propagation between the adjacent blastholes of the cutting seam cartridge under the eccentric and uncoupled situation, the 1/2 model is selected for simulation calculation, the blasthole spacing is 600 mm [45], and other parameters are consistent with the previous article. Use the keyword *MAT_ADD_EROSION* to supplement the definition of failure criteria for rock masses based on the element compressive strength and failure equivalent stress of the rock. In the calculation process, the state of the rock mass is judged. When a certain rock element meets the failure condition, the element is deleted in time to avoid affecting subsequent calculations. The finite element model is shown in Figure 20.

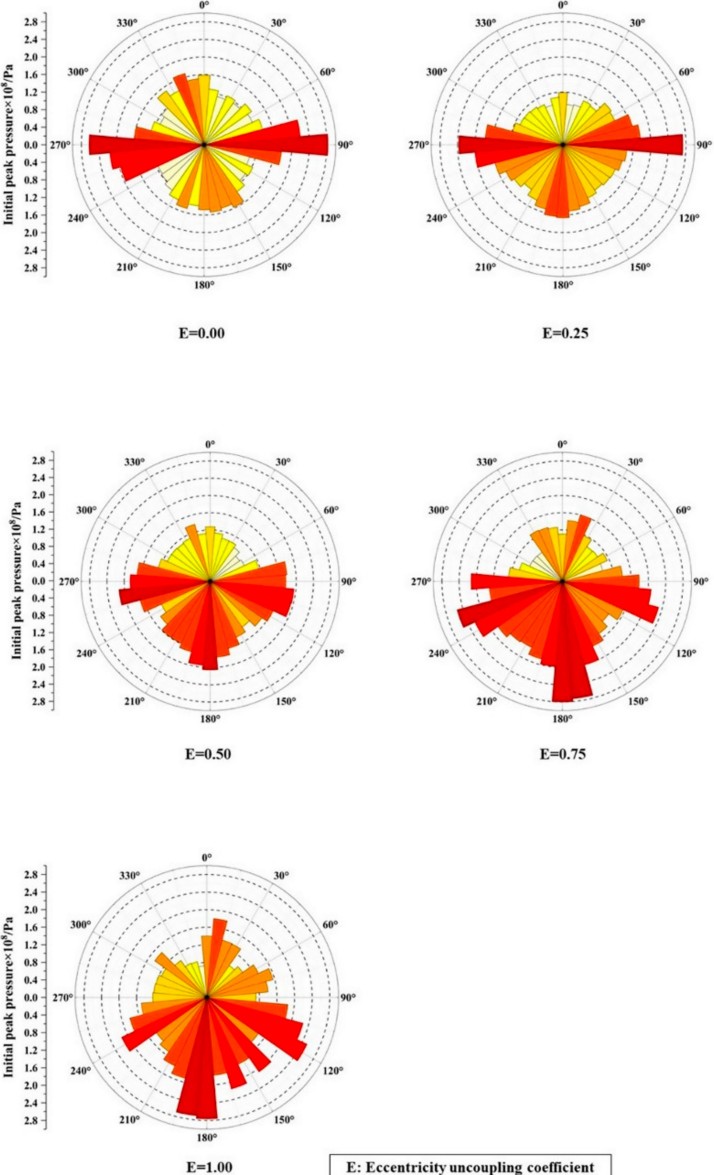

**Figure 19.** Bar graph of the cyclic initial pressure peak.

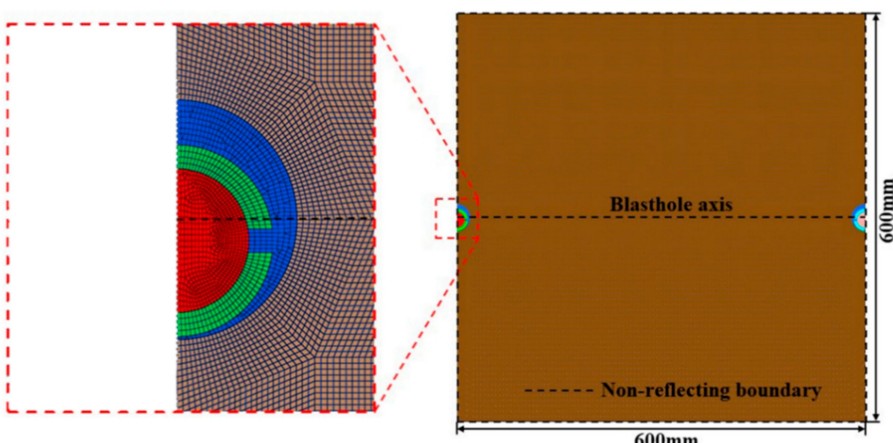

**Figure 20.** Geometry model of double-hole blasting with local magnification mesh.

The cutting seam pipe has a guiding effect on the initial crack. Figure 21 shows the crack propagation under the $E_4$ working condition. In the initial stage of the explosion, the detonation products in the cutting seam direction impacted the rock mass, and the resultant force was greater than the tensile strength of the rock, resulting in cracking and forming several initial cracks. The non-cutting direction has a buffering effect, and only micro-cracks are produced. The $E_4$ working condition is a completely eccentric uncoupled arrangement, the cutting seam pipe on the coupled side is in contact with the blasthole. The initial crack diverges from the axis of the blasthole and expands continuously. The detonation energy is unevenly distributed on the coupled side. The crushing zone and the fracture zone on the coupled side of the blasthole are larger than the uncoupled side, which is quite different from the conclusion of the traditional eccentric uncoupled cartridge. When the cutting seam cartridge is eccentrically arranged without coupling, a small amount of under-excavation will occur at the axis of the blasthole, which will cause less disturbance to the rock mass on the uncoupled side. If in actual blasting, the coupled side is reserved rock mass, the eccentric uncoupled arrangement will cause greater over-excavation and damage.

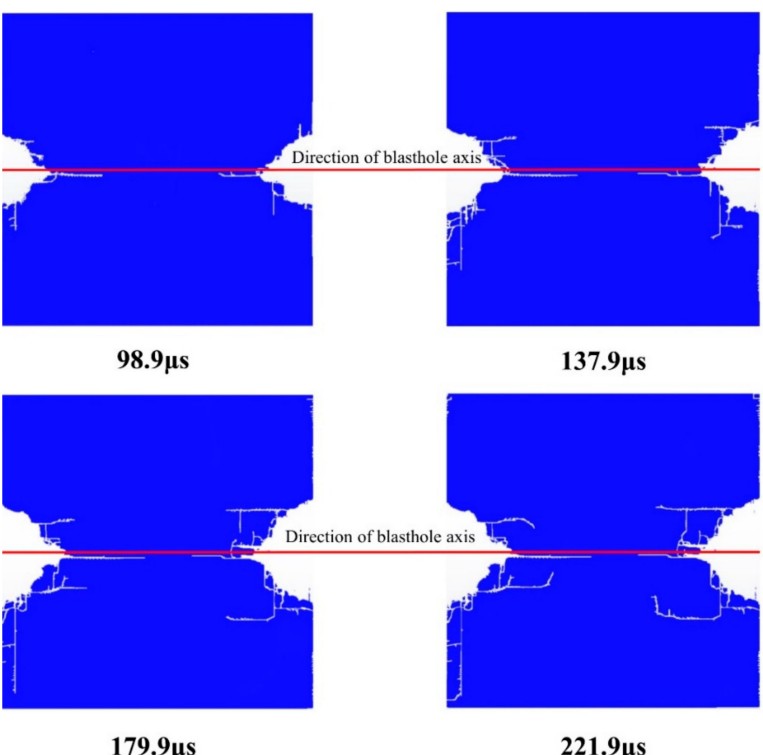

**Figure 21.** Crack propagation under $E_4$ working condition.

## 4. Conclusions

To study the blasting effect and the damage to the rock mass when the cutting seam cartridge is eccentrically and uncoupled. The ANSYS/LS-DYNA® nonlinear dynamic platform was used to simulate the blasting effect of five eccentric uncoupled coefficients on the cutting seam cartridge, and the crack growth process under the condition of complete eccentricity was simulated. By studying the pressure cloud chart, pressure-time interval curve, and blasting effect of the model, the following conclusions are obtained:

1.  When the cutting seam cartridge is blasted, the effect of the detonation product is mainly affected by the wrapping property of the cutting seam pipe and the eccentric uncoupled coefficient. When the eccentric uncoupled coefficient increases from 0 to 1.0, the peak pressure at the connection direction of blasthole decreases by more than 52.7%. This is due to the existence of the cutting seam pipe that causes stress concentration at the cutting seam direction, which enhances the blasting effect in the cutting seam direction and at the same time reduces the damage at the non-

cutting seam direction. And the existence of the cutting seam pipe enhances the guiding effect of the initial crack, and the stress in the non-cutting seam direction is buffered. Therefore, the eccentric arrangement of the cutting seam pipe determines the formation of the initial crack and the subsequent blasting effect;

2. With the increase of the eccentric uncoupled coefficient, the uneven air area in the blasthole has different degrees of buffering and delaying effects on the explosion load, and the load distribution shows obvious unevenness. The occurrence time of pressure peak on the uncoupled side lags behind that on the coupled side. The peak time of coupling side pressure appears earlier with the increase of eccentric uncoupled coefficient. When the eccentric uncoupled coefficient is 1, the peak pressure on the coupled side is 5.78 times that of the uncoupled side. The rock mass damage of the coupled side and uncoupled side is very different, and the explosive stress field is biased toward the coupled side;

3. During the blasting of the cutting seam cartridge, the crushing zone and the fracture zone on the coupled side of the blasthole are larger than that on the uncoupled side, which is quite different from the conclusion when the traditional cartridge is eccentrically arranged. It shows that when the cutting seam cartridge is arranged eccentrically and uncoupled, a small amount of under-excavation will occur at the connection direction of blasthole, and the rock mass disturbance at the uncoupled side will be less. If in actual blasting, the coupled side is reserved rock mass, the eccentric uncoupled arrangement will cause greater over-excavation and damage. Therefore, it is recommended that in the actual blasting construction process, the arrangement of the cutting seam cartridge is ensured to minimize the situation of arranging the coupled side on the retaining side rock mass.

**Author Contributions:** Conceptualization, J.Z. and W.W.; methodology, J.Z.; software, J.Z.; validation, W.W., J.Z. and A.L.; formal analysis, A.L.; investigation, W.W.; resources, W.W.; data curation, A.L.; writing—original draft preparation, J.Z.; writing—review and editing, J.Z.; visualization, A.L.; supervision, W.W.; project administration, W.W.; funding acquisition, W.W. All authors have read and agreed to the published version of the manuscript.

**Funding:** Project (2018JJ2519) supported by the Natural Science Foundation of Hunan Province (51008309); Project (2017041) supported by the Transportation Science and Technology Project of Zhejiang Province; Project supported by the Science and Technology Innovation Program of China Railway Tunnel Bureau Group (Tunnel Research and Integration 2016-14).

**Data Availability Statement:** No new data were created or analyzed in this study. Data sharing is not applicable to this article.

**Conflicts of Interest:** The authors declare no conflict of interest.

## Abbreviations

| | |
|---|---|
| t | Shear stress (Mpa) |
| $S_{ds}$ | Rock dynamic shear strength (Mpa) |
| c | Rock dynamic cohesion (Mpa) |
| $\phi$ | Rock dynamic friction angle (°) |
| $\sigma$ | Maximum cyclic tensile stress (Mpa) |
| $p_d$ | The pressure of the explosion shock wave directly acting on the blasthole wall (Pa) |
| $p_i$ | The pressure of the explosion shock wave acting on the blasthole wall through the cutting seam pipe (Pa) |
| $\mu$ | Poisson's ratio |
| p | Pressure on the blasthole wall (Pa) |
| E | Eccentric uncoupled coefficient |
| $R_1$ | The radius of the blasthole (mm) |

| $R_2$ | The radius of the cutting seam pipe (mm) |
| $OO_2$ | The distance from the center of the cartridge to the center of the blasthole (mm) |
| y | The initial pressure peak (Pa) |
| a | The fitting coefficients |
| b | The fitting coefficients |

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
