# Peer review of "Study on Eccentric Uncoupled Blasting Effect of Cutting Seam Pipe"

_coatings, doi:10.3390/coatings11010104_

Round 1
Reviewer 1 Report
In this paper, based on the LS-DYNA explicit dynamic analysis platform, a quasi-two-dimensional numerical model with five kinds of eccentric uncoupled coefficients is established by using the flour-solid coupling algorithm. Based on the pressure cloud diagram and pressure time history curve output by the post-processing software, the dynamic response behavior characteristics of the explosive, air, the cutting seam pipe, and the rock mass in the eccentric and uncoupled case of the cutting seam cartridge are analyzed. And use the failure element method to simulate the crack propagation effect under completely eccentric conditions. Under the influence of different eccentric uncoupled coefficients, the distribution characteristics of the rock mass explosion stress field and the crack propagation law of the cutting seam cartridge blasting technology are studied to provide a theoretical basis for reducing the possible damage effects.
The paper is quite clear, and the main ideas are quite well developed. I suggest adding some integrations/clarifications prior the publications.
- The main aim of selected methods (advantages/disadvantages) should be presented more clearly.
- Abstract should be expanded by giving more specific results.
- The quality of the figures should be improved to better understanding.
- What are the lessons learned from the conclusions other than the observation? Is this finding original to this paper or is it a validation of a previously established notion/fact? Some philosophical discussions may also be included regarding the findings.
This paper will be useful for academic researchers related to this subject, and for this reason I suggest a review.
Author Response
Dear Editor:
Thank you very much for reconsidering the revised version of our manuscript “Study on eccentric uncoupled blasting effect of cutting seam pipe”. We are thankful for the valuable comments from you and the reviewers.
These comments have been thoughtfully taken into account in the revised manuscript.
The point-by-point responses to the comments are listed on the following pages. All modifications in the revised manuscript are highlighted in cyan-blue color and the responses to reviewers’ comments are highlighted in blue color. The corrected results can refer to the revised manuscript. In the original text, the "track changes" function is
used for modification.
We hope that all these changes fulfill the requirements to make the manuscript acceptable for publication in Coatings. And with best wishes for a happy New Year!
Sincerely yours,
Jiaqi Zhang

Reviewer 2 Report
Congratulations. The work is good, however, there are some concerns about your work, which can be addressed, improving the paper and its understanding. Please see attachment.

Author Response

(The authors gave the same response as above.)

Reviewer 3 Report
The paper called Study on eccentric uncoupled blasting effect of 2 cutting seam pipe by Jiaqi Zhang, Wei Wang and Ao Liu, is quite original. It deals with cutting technology. The paper should present some more references dealing with an overall perspective and possible applications of presented technology. English is good.
The paper is well written but there are few things that should be changed:
- The abstract should present the significance of the paper in more detail and the presentation of the methods, analysis and findings should be expanded.
- The main goal and scope of the research and the paper itself should be highlighted.
- Please, elaborate on the results in more detail.
- Lines 7-12 – review the style and grammar.
- Lines 152-153 – the sentence is cut in half.
- Lines 197 and 415 – remove the dot.
- Lines 430-432 – review the style and grammar. Maybe start with “the blasting effect of the…”
The presented conclusions have potential; therefore, they should be presented in a better light and the authors should emphasize the original research contribution. Suggested amendments will significantly increase the relevance of the publication and will improve it. After applying all the required changes, the paper is suitable for publication.
Author Response

(The authors gave the same response as above.)

Round 2
Reviewer 1 Report
The paper is now accepted as is